# A Re-Examination of Neoadjuvant Therapy for Thymic Tumors: A Long and Winding Road

**DOI:** 10.3390/cancers16091680

**Published:** 2024-04-26

**Authors:** Fenghao Yu, Zhitao Gu, Xuefei Zhang, Ning Xu, Xiuxiu Hao, Changlu Wang, Yizhuo Zhao, Teng Mao, Wentao Fang

**Affiliations:** 1Department of Thoracic Surgery, Shanghai Chest Hospital, Shanghai Jiao Tong University School of Medicine, Shanghai 200030, China; yfh19961224@yahoo.com (F.Y.); guzhitao0717@hotmail.com (Z.G.); docchang@126.com (X.Z.); mouse0326@163.com (N.X.); x2kiara@gmail.com (X.H.); hippomao@hotmail.com (T.M.); 2Department of Radiation Oncology, Shanghai Chest Hospital, Shanghai Jiao Tong University School of Medicine, Shanghai 200030, China; luise2w@msn.com; 3Department of Pulmonary Medicine, Shanghai Chest Hospital, Shanghai Jiao Tong University School of Medicine, Shanghai 200030, China; zhyz920916@126.com

**Keywords:** thymic epithelial tumors, neoadjuvant therapy

## Abstract

**Simple Summary:**

Thymic epithelial tumors are a rare and mostly indolent type of solid tumors. Histology, tumor staging, and resection status are important indicators of prognosis. Preoperative therapy is utilized in patients with advanced thymic tumors when primary surgery is unlikely to achieve R0 resection. By administering preoperative therapy, clinicians hope to have a better chance of resecting the tumor completely so that the patient may have a better prognosis. Many retrospective and a few prospective studies have been conducted in the past two decades on preoperative therapy. Also, many novel agents have recently entered the field of thymic tumor treatment. This review is a re-examination of the current evidence.

**Abstract:**

For most patients with advanced thymic epithelial tumors (TETs), a complete resection is a strong indicator of a better prognosis. But sometimes, primary surgery is unsatisfactory, and preoperative therapy is needed to facilitate complete resection. Neoadjuvant chemotherapy is the most used form of preoperative therapy. But studies on neoadjuvant chemotherapy have included mainly patients with thymoma; its efficacy in patients with thymic carcinoma is less known. Neoadjuvant chemoradiation has also been explored in a few studies. Novel therapies such as immunotherapy and targeted therapy have shown efficacy in patients with recurrent/metastatic TETs as a second-line option; their role as preoperative therapy is still under investigation. In this review, we discuss the existing evidence on preoperative therapy and the insight it provides for current clinical practice and future studies.

## 1. Introduction

The treatment of thymic epithelial tumors (TETs) gravitates heavily towards their resectability and the possibility of complete resection. While the management of early-stage TETs progresses on a straightforward path with surgical resection as the main treatment modality, the management of locally advanced TETs is more complicated. To achieve a complete resection is still the goal for most locally advanced cases, as R0 resection is a stronger indicator for better prognosis. But results of a primary surgery sometimes prove to be unsatisfactory when tumor invasion is extensive and the probability for complete resection is low. In these cases, neoadjuvant therapy comes in to bridge the gap [1,2].

Neoadjuvant chemotherapy is most commonly used in today’s practice, and neoadjuvant chemoradiation also shows some efficacy. However, due to the rarity of TETs, most studies on preoperative therapy were retrospective, without a control group, and with small sample sizes [3,4,5,6,7,8,9]; no randomized controlled trial currently exists, which is typical for a rare tumor like TET. In addition, the understanding of TET biology and histology has evolved greatly in the past two decades. Inclusion criteria in terms of tumor resectability and histology vary significantly among previous studies, making direct comparison of their results difficult. And with novel agents of immunotherapy and targeted therapy entering the field of TET treatment, clinical trials on them as preoperative treatment options are underway.

This article focuses on evidence, old and new, on preoperative therapy for TETs and discusses how well results from these studies, with their strengths and weaknesses, are translated into current practice. In addition, this article looks into promising novel approaches and discusses potential options for the future.

## 2. The Starting Point for Neoadjuvant Therapy

Determining the resectability of a TET and whether complete resection is possible is no easy task. Decision making is based on the clinical stage (predominantly using the Masaoka–Koga staging system) and relies heavily on the interpretation of pre-operative imaging and surgeons’ experience. A tumor might be ‘unresectable’ in one surgeon’s opinion and yet ‘resectable’ in another’s. Expressions like ‘judged to be unresectable’ were heavily used in clinical trials for lack of an objective standard [3,5].

Currently, two staging systems are most commonly employed in daily practices and clinical trials. One is the Masaoka–Koga staging system [10]; the other is the Union for International Cancer Control/American Joint Committee on Cancer (UICC/AJCC) TNM staging system [11]. The former was built more than 30 years ago based on a very limited number of cases from single-institution data, and studies published in the 2000s and 2010s mainly used this system in their inclusion/exclusion criteria [3,5]. While stage I–II disease may be readily resectable, stage III is defined as an invasion into any neighboring organs, regardless of their resectability [12]. Compared to the Masaoka–Koga staging system, the eighth TNM staging system, based on a much larger global database, was more refined and less heterogeneous in terms of tumor resectability. T4 structures of the myocardium, aorta and its branches, intra-pericardial pulmonary vessels, trachea, and esophagus are generally considered unresectable, differentiating them from potentially resectable T3 structures. The TNM staging system, therefore, is more helpful in determining resectability. But T3 structure are still composed of a variety of neighboring structures of different levels of resectability [13]. An invasion of the phrenic nerve and limited lung parenchyma (T3 based on the eighth TNM staging system) is amenable to upfront surgery and might even be completely resected by minimally invasive thymectomy (MIT) in experienced hands [14]. The reclassification of the phrenic nerve and lung parenchyma in the forthcoming ninth edition of the TNM staging system as T2 structures also indicates the difference of resectability [15]. The rest of the T3 (eighth TNM) structures are more complicated to assign in terms of resectability, especially when more than one structure is involved. Stage alone is sometimes a poor indicator of resectability even when the TNM staging system is employed because heterogeneity still exists within the same T, N, and M category. The concept of resection index (RI) recently suggested by Gu et al. proposed a more detailed measurement of technical difficulty in addition to the T category [14]. The pericardium was assigned a score of 1, phrenic nerve and lungs a score of 2, and vascular structures such as innominate veins a score of 3, in order to reflect the difficulty in their resection. The final RI, which was calculated according to the total number of invaded structures and their difficulty scores, would be a better way to represent tumor resectability. Therefore, instead of the two polar opposites of ‘resectable’ and ‘unresectable’, there could be a spectrum of resectability. Most clinical cases would fall into the middle of ‘potentially resectable’, in which upfront surgery could be attempted, but R0 might be compromised or with greater peri-operative risks.

One of the reasons that makes a direct comparison of results from different trials difficult is the drastic difference observed among studies when the word ‘unresectable’ is used. For instance, in a phase II, multi-institutional trial on neoadjuvant chemoradiation for advanced TETs [7], a specific scheme based on CT features was used for patient accrual, and tumors > 8 cm in the greatest axial diameter alone were one of the inclusion criteria. This may come from the rationale that, although not directly associated with invasiveness, tumor size is believed to affect resectability and prognosis [16,17]. Other studies simply rely on clinical stage and clinician judgement.

Moreover, histology also has a part to play when choosing the optimal treatment modality. Thymoma and thymic carcinoma represent two distinctive histology subtypes. They have different genetic profiles and run different clinical courses [16,17,18]. Different chemotherapy regimens result in different responses from thymomas and thymic carcinomas. The results from clinical trials implied that different histological types might respond differently to preoperative chemoradiation, although the evidence was not strong. Historically, what had been validated in thymoma management was deemed applicable in thymic carcinoma management—for example, the Masaoka–Koga staging system. In addition, the histological classification of TETs has also evolved, causing reporting terminology to change. For example, a term like ‘well-differentiated thymic carcinoma’ was used to refer to a type B3 thymoma in the second edition of the WHO’s histological classification. Trials in which patients from the 90s were accrued used ‘lymphocytic’, ‘mixed’, and ‘epithelial’ to refer to thymoma subtypes. Therefore, it is important for clinicians to be aware of the discrepancy among criteria used in different studies when interpreting their results.

## 3. Neoadjuvant Chemotherapy—A Familiar Road to Nowhere?

The ultimate goal of neoadjuvant treatment is to improve the disease-free survival of patients through transforming ‘unresectable’ tumors to become ‘resectable’ and facilitating R0 resection. The existing evidence, with a few exceptions, has failed to demonstrate the direct survival benefits of preoperative therapy in advanced TETs [8,9,19,20,21]. The R0 rate, along with ORR, is therefore favored as a surrogate endpoint, indicating good results.

Chemotherapy has long been accepted into the treatment of TETs. Response rates vary among regimens, with some trials reporting ORR over 90% [22,23,24]. Neoadjuvant chemotherapy became the recommended approach for unresectable TETs in the guidelines based on the results of two single-arm small-sample phase II trials conducted almost two decades ago [1,2] (See Table 1). How well these old results can be translated into today’s practice, however, needs to be carefully re-examined. Both trials only included patients with unresectable thymoma. Different first-line chemotherapy regimens are recommended for thymoma and thymic carcinoma, such as cyclophosphamide, doxorubicin, and cisplatin (CAP) for thymoma and paclitaxel and carboplatin for thymic carcinoma [2]. But a great deal of crossover is present in previous and ongoing studies, especially in periods when pathological classification and reporting terminology were less standardized [21,25]. It is natural to ask whether the treatment strategy of neoadjuvant chemotherapy plus surgery is as effective for patients with thymic carcinoma as for patients with thymoma. In Kim’s and Kunitoh’s trials for patients with potentially unresectable thymomas, the reported ORRs were 77% and 62%, and R0 rates were 76% and 69%, respectively, in patients proceeding to surgery [3,5]. Both trials defined partial response as a 50% or greater reduction in tumor diameter. Six patients had tumors of >80% necrosis in Kim’s study, and three patients in Kunitoh’s trial had pathological complete responses (pCRs). In comparison, very little evidence supports neoadjuvant chemotherapy for patients with thymic carcinoma. In a prospective phase II trial of neoadjuvant chemotherapy for unresectable thymoma and thymic carcinoma by Park et al. [26], the results seemed promising: out of 16 thymic carcinoma patients (excluding two NETTs in the original group), there were 12 partial responses (PRs) and four stable diseases (SDs). However, its results need to be considered and evaluated against the backdrop of a rather moderate response of thymic carcinoma to palliative chemotherapy. A multi-center phase II study of carboplatin and paclitaxel for patients with advanced thymic carcinoma (not amenable to curative surgery or radiotherapy) by Hirai et al. showed only 36% ORR in 36 thymic carcinoma patients [27]. The results of the phase II study by Lemma et al. showed that compared to advanced thymoma patients (*n* = 21), advanced thymic carcinoma patients (*n* = 23) seemed to have a worse response to paclitaxel and carboplatin (ORR: 42.9% vs. 21.7%) [25]. Contradictory results exist when comparing thymoma and thymic carcinoma’s responses to the same regimen [28,29]. According to the results from the retrospective NEJ023 study, the efficacies of different first-line regimens did not vary significantly in patients with advanced thymic carcinoma in terms of response. Multidrug chemotherapy regimens seemed to result in similar ORR to that of platinum-based doublets (42.7% vs. 38.2%) but were associated with higher toxicity [30]. Therefore, neoadjuvant chemotherapy, especially for patients with thymic carcinoma, needs to be further assessed as to whether or not it can result in an optimal ORR that could be translated to higher resectability and better prognosis. If not, other treatment options such as combining different modalities or adding novel agents to existing regimens need to be explored in future studies.

The phase II trial by Kim et al. [5] also had a very loose definition of unresectable thymoma by today’s standards, as shown by the descriptions of tumor invasion. For instance, tumors with pleura invasion were also included as potentially unresectable based on pre-operative imaging. The purpose of preoperative therapy might be defeated when R0 resection is easily achieved by upfront surgery. By comparison, the inclusion criteria in the phase II trial by Kunitoh et al. [3] were more stringent in terms of tumor resectability—invasion into the pulmonary artery trunk in 10 cases, superior vena cava in 8 cases, aorta in 6 cases, extensive pericardium or myocardium in 4 cases, and sternum in 1 case. In addition, the regimen used by Kim et al. contained a high dose of prednisone. Corticosteroids have been shown to induce tumor regression in lymphocyte-rich thymomas [31,32], and 45% of the tumors included in that trial were described as lymphocytic. Thus, it was logical that a much lower R0 rate in the intention-to-treat population was present in the latter trial, therefore resulting in much lower progression-free survival at 5 years (77% in Kim’s study and 43% in Kunitoh’s). However, no survival benefit was found in patients who underwent surgical resection over those who did not in Kunitoh’s study, probably due to its small sample size. The divergence on resectability again calls for a more objective method to define the extent of tumor invasion. And the aforementioned concept of RI might be a good starting point [14].

## 4. Neoadjuvant Chemoradiation—1 + 1 > 2?

Similar to chemotherapy, the role of radiotherapy is well established in the treatment of TETs. Radiotherapy is widely used in the adjuvant setting and has been shown to reduce tumor recurrence after complete resection, although controversy exists on the optimal circumstance under which it should be administered [2,33,34]. Concurrent chemoradiation is recommended as the definitive treatment for TETs when surgery is no longer an option [1,2]. However, limited evidence is available on the use of chemoradiation in the neoadjuvant setting. In a retrospective study by Wright et al. [35], ten patients with unresectable TETs were treated with concurrent chemoradiation (two cycles of etoposide and cisplatin plus radiation of 40–50 Gy as the target dosage). Only one patient with thymic carcinoma was included. All patients completed preoperative therapy. No disease progression was observed, and 40% of the patients had a partial response. R0 resection was achieved in 8 out of the 10 patients. Two patients had a near pathological complete response (necrosis > 99%), and another two had very limited residual disease (necrosis > 90%). The regimen was well tolerated, with no notable toxicity or death observed during treatment. A following phase II trial [7] with the same induction regimen reported an ORR of 48% and an R0 rate of 80.9% in patients who underwent surgical resection. Seven patients with thymic carcinoma were included. Grade 3 and 4 toxicities from preoperative therapy occurred in 9 out of 21 patients. What is interesting about this trial is that instead of only including unresectable TETs, tumors of both advanced stages and large sizes were included. Even though no patient had a complete pathologic response, five specimens of thymic carcinoma (24%) had <10% viable tumor.

Chemoradiation shows promising results as a neoadjuvant therapy in terms of ORR and the R0 rate. Whether a near pathological complete response as reported by Wright et al. and Korst et al. could be translated into survival benefits remains to be seen. However, several aspects need to be considered. First, patients with thymic carcinoma were still under-represented (one in Wright’s study and seven in Korst’s). In Korst’s study, 3 out of 7 patients with thymic carcinoma reached PR, compared with 7 out of 14 patients with thymoma. Second, no direct evidence favors neoadjuvant chemotherapy or neoadjuvant chemoradiation over the other, in terms of better response, higher R0 rate, or better survival. The efficacy of neoadjuvant concurrent chemoradiation may be non-inferior to that of neoadjuvant chemotherapy. But many factors may hinder direct comparison, including different inclusion criteria and different radiographic response evaluation criteria. The 1 + 1 > 2 effect of neoadjuvant chemoradiation for TETs is for now largely hypothetical. Finally, even though no severe toxicity was conspicuous in Korst’s study, toxicity from concurrent chemoradiation in theory might hinder its implementation. Therefore, while more evidence on the efficacy of concurrent chemoradiation needs to be gathered, the role of neoadjuvant sequential chemoradiation also deserves investigation. A recent retrospective study comparing neoadjuvant concurrent and sequential chemoradiation showed similar responses (80.0% and 62.5%, *p* > 0.05) and R0 rates (80.0 and 68.8%, *p* > 0.05), with a lower probability of hematologic adverse events in sequential chemoradiation (*p* = 0.009) [36]. Additionally, sequential chemoradiation, in theory, might also save some patients from unnecessary preoperative radiation if chemotherapy alone could result in satisfactory response. The results from ongoing trials on concurrent and sequential chemoradiation as preoperative therapies are much expected so that clinical decision could be further optimized in the future.

## 5. Novel Agents—New Horizon in Near Sight

Promising novel agents may provide new opportunities for preoperative therapy. Immune checkpoint inhibitors (ICIs) have revolutionized cancer treatment and brought hope for patients with thoracic malignancies, including lung cancer and esophageal cancer. It now covers clinical scenarios from palliative to neo-adjuvant and adjuvant therapies for resectable diseases [37,38,39]. This new landscape has also expanded to include thymic epithelial tumors, as high levels of PD-L1 expression have been found in thymoma and thymic carcinoma [40].

An open-label, phase II trial using pembrolizumab on refractory or relapsed TETs reported a 28.6% PR rate in patients with thymoma and 19.2% in those with thymic carcinoma. However, the rate of immune-related adverse events (irAEs) was dangerously high in thymoma patients [41]. Five of seven patients (71.4%) with thymoma experienced grade ≥ 3 irAEs (15.4% in patients with thymic carcinoma). In another phase II trial, second-line pembrolizumab produced a 22.5% ORR in 40 patients with recurrent thymic carcinoma, including one complete response, eight partial responses, and 21 stable diseases. Only 15% of the treated patients developed severe autoimmune toxicity [42]. In a multi-cohort phase II trial of atezolizumab in patients with solid tumors whose disease progressed after one or more lines of systemic therapy, a 38.5% ORR was seen in 13 patients with advanced thymoma, but similarly with pembrolizumab, an alarmingly high rate of serious adverse events (35.7%) and one treatment-related death were reported [43]. The combination of avelumab, an anti-PD-L1 inhibitor, with axitinib, an anti-angiogenesis drug also showed promising results in the second-line setting. In 32 unresectable or metastatic B3 thymoma and thymic carcinoma patients, 11 had an overall response rate (ORR = 34%) [44]. This study suggests there might be potential synergistic effects by combining novel agents of different mechanisms in the treatment of TETs. Currently, several trials on the combination of immunotherapy and tyrosine kinase inhibitor are underway (NCT03463460, NCT04710628), and their results are highly expected.

Based on the promising efficacy and relative safety of ICIs in the second-line treatment of patients with thymic carcinoma, more robust responses might be expected as they enter the first-line setting with or without chemotherapy. A successful case of chemotherapy plus pembrolizumab as neoadjuvant therapy has been reported [45]. Several registered phase II trials are now underway investigating the efficacy and safety of different PD-1 and PD-L1 inhibitors (see Table 2), not only as the first-line treatment for unresectable/metastatic TETs (NCT04554524, NCT05832827, ChiCTR2300072705) but also as a neoadjuvant treatment for advanced TETs (NCT03858582, NCT04667793, NCT06019468, ChiCTR2300074152). It is notable that most of these ongoing trials have chosen immunotherapy plus chemotherapy rather than immunotherapy alone.

Several targeted therapy agents also showed a certain efficacy in TET treatment. Some of them could be potential candidates as preoperative therapy for their ability to bring about noticeable tumor regression. Unlike in the case of lung cancers, however, targetable gene mutation is a rare event in thymic tumors [46,47]. Sunitinib is an anti-angiogenic multi-kinase inhibitor targeting VEGFR, PDGFR, and c-KIT. A phase II trial on sunitinib in patients with chemotherapy-refractory thymoma and thymic carcinoma reported a 26% ORR in thymic carcinoma patients and a 6% ORR in thymoma patients. Increases in Treg PD-1 expression and CD8+ T-cell CTLA4 expression were also noted. [48]. Therefore, adding ICI to sunitinib might result in a greater response. A more recent phase II trial confirmed sunitinib’s efficacy in pre-treated thymic carcinomas with a reported ORR of 21.7% [49]. Lenvatinib, an oral multi-kinase inhibitor that targets VEGFR, FGFR, c-KIT, and other kinases, produced a 36% ORR as a second-line therapy for unresectable advanced or metastatic thymic carcinoma patients [50]. Some small molecule anti-angiogenic drugs also showed promising results. Retrospective studies showed that the ORR of anlotinib and apatinib was around 20–40% in the second-line setting [51,52,53]. In a phase II trial, second-line apatinib produced an ORR of 40% in patients with recurrent or metastatic TETs [54]. Ongoing trials are investigating different targeted therapies in the first-line setting. The REVELENT trial is a multicenter, open-label, single-arm, phase II study aimed at investigating the activity and safety of ramucirumab combined with paclitaxel and carboplatin in chemotherapy-naive patients affected by thymic carcinoma or B3 thymoma with an area of carcinoma [55]. A phase II trial of chemotherapy plus cetuximab followed by surgical resection in TETs is now underway, investigating the efficacy of targeted therapy plus chemotherapy in the neoadjuvant setting (NCT01025089), as EGFR overexpression was common in TETs [56].

As mentioned before, corticosteroids alone have been reported to induce tumor regression [31,32]. Kobayashi et al. reported using steroid pulse therapy (intravenous infusion of 1 g of methylprednisolone each day for 3 days) as a preoperative treatment for patients with invasive thymoma. No severe toxicity or complication occurred during the peri-operative period. The ORR was 47.1% in 17 patients [57]. The response was only observed in lymphocyte-rich thymomas, and tumor size reduction was most prominent in B1 thymomas. Although showing promising efficacy, the role of corticosteroids as a preoperative therapy needs to be further investigated in prospective trials, as well as its mechanism for anti-tumor effects in TETs. This old agent may bring new life into the management dilemma when orthodox approaches such as chemotherapy or chemoradiation fail to bring satisfactory results.

Although they are at a very early stage, these promising results from novel agents may lead to a new horizon. A robust anti-tumor effect is anticipated for future preoperative therapy augmented by novel agents. Viable biomarkers are needed to better guide patient selection. PD-L1 expression and tumor mutational burden (TMB) are well-known predictors of response to immunotherapy in other tumors. The former is common in TETs, but the latter has been reported to be very low (an average of 0.48 mutations per megabase) [16,58]. A relatively high response to immunotherapy was observed in the treatment of TETs in spite of low TMB. The expression of PDL1 and alterations in genes or pathways that are correlated with PD-L1 expression could potentially predict the response to immunotherapy in patients with advanced thymic carcinoma [59]. Safety is also a major concern for patients with TETs treated with ICIs. Patient selection based on histological subtypes needs to be performed with extra caution so as to avoid severe irAEs. It is important to understand the mechanism behind immune-mediated toxicity and the unique immune landscape of TETs to provide a safe and effective treatment [60]. As for targeted therapy, the quest for a more comprehensive view of TETs’ molecular and genetic profile is still ongoing. A GTF2I mutation has been found repeatedly in certain subtypes of thymoma [61,62] and a GTF2I knock-in mouse model of thymoma has been successfully developed [63]. However, it is currently untargetable. Other known targetable mutations are rare in TETs, and this remains the key obstacle to more effective targeted therapy.

## 6. Conclusions

For a rare disease such as TET, it is difficult to gather high-quality evidence from RCTs. The retrospective and prospective studies mentioned above have set the tone for preoperative therapy in today’s clinical practices. However, it is still too early in the journey to favor one way over the other. Current evidence suggests that differences in response to preoperative therapy exist between thymomas and thymic carcinoma. Therefore, they should be regarded as different entities when investigating options for preoperative therapy. In addition to this, a more objective definition of resectability in addition to staging is much needed to guide treatment strategy.

There still exists a great need to search for treatment modality to further promote tumor down-staging and facilitate complete resection. While neoadjuvant chemotherapy is currently the best option for patients with thymoma, the role of neoadjuvant chemotherapy needs to be further investigated in patients with thymic carcinoma. More evidence needs to be accumulated on concurrent and sequential chemoradiation, their respective safety, and their efficacy. The role of immunotherapy and targeted therapy in the neoadjuvant setting awaits results from ongoing trials. The breaking dawn of preoperative therapy and hope for patients with locally advanced TETs (especially locally advanced thymic carcinoma) in the next decade might be brought by them. But before this, a more comprehensive view of TET biology is needed to better guide study design in the future.

## Figures and Tables

**Table 1 cancers-16-01680-t001:** Prospective trials on preoperative therapy in TETs.

Study	Eligible Patient Number	TM/TC	Preoperative Modality	ORR	R0 Rate in Resection Patients	Pathological Response
Kim 2004 [5]	22	22/0	Chemo: cyclophosphamide, doxorubicin, cisplatin, prednisone	77.3%	76.2% (16/21)	6 major responses (>80% necrosis)
Kunitoh 2010 [3]	21	23/0	Chemo: vincristine, doxorubicin, cisplatin	61.9%	69.2% (9/13)	3 pCRs
Park 2013 [26]	27	9/18	Chemo: docetaxel, cisplatin	63.0%	78.9% (15/19)	Not mentioned
Korst 2014 [7]	21	14/7	Chemo: etoposide, cisplatin Concurrent radiation: ≤45 Gy	47.6%	80.9 (17/21)	5 major responses (<10% viable tumor)

**Table 2 cancers-16-01680-t002:** Ongoing phase II trials of first-line immunotherapy in TET treatment.

Trial Identifier	Setting/Patients	Regimen	Instituion
NCT04554524	First-line/Unresectable TETs	Paclitaxel + Pembrolizumab	Tangdu Hospital, China
NCT05832827	First-line/Unresectable TCs	Paclitaxel + Carboplatin + Lenvatinib + Pembrolizumab	National Cancer Center, Japan
ChiCTR2300072705	First-line/Unresectable TCs	Paclitaxel + Crboplatin + Adebrelimab	Shanghai Chest Hospital, China
NCT03858582	Neoadjuvant first-line/Advanced TETs	Docetaxel + Cisplatin + Pembrolizumab	Samsung Medical Center, Republic of Korea
NCT04667793	Neoadjuvant first-line/Advanced TETs	TMs:Cisplatin + Amycin + Cyclophosphamide + Toripalimab TCs: Carboplatin + Paclitaxel + Toripalimab	Shanghai Pulmonary Hospital, China
NCT06019468	Neoadjuvant first-line/Advanced TCs	Envolizumab + Radiotherapy	Shanghai Pulmonary Hospital, China
ChiCTR2300074152	Neoadjuvant first-line/Advanced TCs	Paclitaxel + Crboplatin + Adebrelimab	Shanghai Chest Hospital, China

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
