# Peer review of "A Re-Examination of Neoadjuvant Therapy for Thymic Tumors: A Long and Winding Road"

_cancers, 2024, doi:10.3390/cancers16091680_

Round 1

Reviewer 1 Report

Comments and Suggestions for Authors

This paper is exhaustive for the objective and future perspectives of the induction therapy in thymic tumors.

Comments on the Quality of English Language

English language is good. Need a minor revision.

Author Response

Thank you for your affirmation of the content of our manuscript. Revision of the language has been made.

Reviewer 2 Report

Comments and Suggestions for Authors

The manuscript is well designed and well structured in sections.

Unfortunately, the topic addressed is an area lacking of evidence, with very few available and completed trials, with an inadequate number of patients enrolled and – as a result – with poor evidence.

Results and conclusions are controversial in this area.

Many biases are presents and make interpretation misleading.

I feel this is the main problem with the manuscript, the evidence of what discussed is limited.

While this is clearly stated by the Authors in the title (long and winding road), as a consequence the topic is of reduced interest in my opinion and less appealing.

To make the manuscript more attractive I would completely reengineer the structure of it: less space dedicated to review results/role of induction CT or CT/RT (currently the presentation ends at line 254), where the evidence is poor. I will stress more the bias of the available studies (carcinoma histology vs. thymomas, concept of resecability/unresecability, stages included, pleural disease, etc). Stating in few lines the current evidence.

On the contrary I would dedicate much more space on the ongoing trials with immunotherapy, preliminary results, expected benefit, rationale in thymic tumours.

Overall it is in my opinion a well written article that deserves publication after some revision

Author Response

Thank you for your comment. We agree, as you mentioned, that more evidence is needed for a definitive answer in this area. Therefore, according to your advice, we streamlined the structure of our manuscript mainly in the induction chemotherapy part to focus more on the bias of the available studies, highlighting histology and resectability. Also, more content was added in the novel agent part to focus on their potential roles as induction therapy.

Reviewer 3 Report

Comments and Suggestions for Authors

nice review.

Comments on the Quality of English Language

Nice review but I believe that AI was used from the authors 

Author Response

Thank you for your affirmation. No AI was used in the writing of our manuscript.

Reviewer 4 Report

Comments and Suggestions for Authors

This review article entitled "Re-examination of induction therapy for thymic tumors: a long and winding road (cancers-2884216)" by Dr. Yu et al. describes perioperative chemotherapy with a focus on induction chemotherapy for thymic epithelial tumours. I am afraid that this review article should be improved because it is not comprehensive and contains high insight.

Main points.

1. I recommend the authors to update the latest results of clinical trials.

2. Include a clinical trial of anti-angiogenic agents for thymic epithelial tumours.

3. Reconstruct the division between thymoma and thymic carcinoma. They are quite different in terms of genetic alteration (or biological characteristics).

4. The clinical studies included are different from the WHO pathological classification. The identical criteria of thymoma and thymic carcinoma have changed in the old classification. I advised the authors to be aware of this.

5. I recommend the authors to read the ITMIG guidelines including terminology,

6. The authors to describe the concrete numbers, not ambiguous expression.

7. In the conclusion section, the description is long and redundant, as well as no insights.

Minor

1. Unify the technical terms, P1L15: upfront surgery will be appropriate to be primary surgery, also "difficult" is ambiguous.

2. P1L24-27: I cannot understand here. Some ambiguous terminology or expressions should be revised.

3. P2L63-64: I cannot understand well here. Some ambiguous terms or expressions should be revised.

4. P2L69: Does "potentially resectable" mean "resectable"?

5. P4L157-159: These are not precise. The concepts of "thymoma" and "thymic carcinoma" have changed over time. The authors should also clarify the main drugs for thymoma and thymic carcinoma.

6. P4L173-174: NEJ023 is a retrospective study. The authors should not mix phase 2 trials and retrospective studies.

7. P8L287-288: The authors consider lenvatinib and sunitinib the same. I disagree. The response rate and the PD rate of both drugs are different.

8. P8L296: I recommend that the authors rephrase here.

9. P8L319-333: The expression "very low" is ambiguous. The reader will want to know the number.

Comments on the Quality of English Language

Moderate change will be required, but I am afraid it is not always a matter of expression.

Author Response

Thank you for your comments. We provide a point by point response below.

Major point responses

  1. Latest results have been included in novel agent section. See P6L257-280.
  2. Several trials on apatinib and anlotinib as well as the mentioning of the ongoing REVELENT trial were added. See P6L257-280.
  3.  We have revised as advised. See the neoadjuvant chemotherapy section as well as P3L104-117
  4.  We have added several statements. See P3L104-117.
  5.  We have changed induction to the more recommended preoperative and neoadjuvant.
  6. We have revised accordingly throughout the manuscript.
  7. The conclusion section has been simplified and reconstructed and hopefully it contains more insights now.

Minor point responses

  1. Upfront surgery is changed to primary surgery. Difficult is specified as unlikely to achieve R0 resection.
  2. Original expression is now replaced by "Neoadjuvant chemotherapy is the most used form of preoperative therapy. But studies on neoadjuvant chemotherapy included mainly patients with thymoma; its efficacy in patients with thymic carcinoma is less known".
  3.  The original line is deleted for it was redundant.
  4.  Potentially resectable was explained in the following sentence: upfront surgery could be attempted but R0 resection might be compromised or with greater peri-operative risks. From a surgeon's point of view, we feel like there is a subtle difference among resectable, potentially resectable and unresectable. We are aware that TNM staging is recommended over vague terms. But the point we would like to make (by extensive efforts in our manuscript) is that staging alone is inadequate in describing TET's resectability. For revised statement see P3L92-95.
  5. We reconstructed the neoadjuvant chemotherapy part in our manuscript to highlight the difference of thymoma and thymic carcinoma.
  6. The word retrospective was added. Mentioning this study was to further highlight the uncertainty of chemotherapy's efficacy in patients with thymic carcinoma.
  7. Expressions were changed to avoid confusion. Though they were both prospective trials, patient size was small in both studies. We wanted to avoid reading too much into their results by comparing efficacies.
  8. The expression has been rephrased.
  9. "Average of 0.48 mutations per megabase" was added.